# Constructing an atlas of associations between polygenic scores from across the human phenome and circulating metabolic biomarkers

Si Fang*, Michael V Holmes, Tom R Gaunt, George Davey Smith, Tom G Richardson*

MRC Integrative Epidemiology Unit (IEU), Population Health Sciences, Bristol Medical School, University of Bristol, Bristol, United Kingdom

## Abstract

**Background:** Polygenic scores (PGS) are becoming an increasingly popular approach to predict complex disease risk, although they also hold the potential to develop insight into the molecular profiles of patients with an elevated genetic predisposition to disease.

**Methods:** We sought to construct an atlas of associations between 125 different PGS derived using results from genome-wide association studies and 249 circulating metabolites in up to 83,004 participants from the UK Biobank.

**Results:** As an exemplar to demonstrate the value of this atlas, we conducted a hypothesis-free evaluation of all associations with glycoprotein acetyls (GlycA), an inflammatory biomarker. Using bidirectional Mendelian randomization, we find that the associations highlighted likely reflect the effect of risk factors, such as adiposity or liability towards smoking, on systemic inflammation as opposed to the converse direction. Moreover, we repeated all analyses in our atlas within age strata to investigate potential sources of collider bias, such as medication usage. This was exemplified by comparing associations between lipoprotein lipid profiles and the coronary artery disease PGS in the youngest and oldest age strata, which had differing proportions of individuals undergoing statin therapy. Lastly, we generated all PGS–metabolite associations stratified by sex and separately after excluding 13 established lipid-associated loci to further evaluate the robustness of findings.

**Conclusions:** We envisage that the atlas of results constructed in our study will motivate future hypothesis generation and help prioritize and deprioritize circulating metabolic traits for in-depth investigations. All results can be visualized and downloaded at http://mrcieu.mrsoftware.org/metabolites_PRS_atlas.

**Funding:** This work is supported by funding from the Wellcome Trust, the British Heart Foundation, and the Medical Research Council Integrative Epidemiology Unit.

**\*For correspondence:**
si.fang@bristol.ac.uk (SF);
Tom.G.Richardson@bristol.ac.uk (TGR)

## Editor's evaluation

The authors describe their work on an atlas of associations between polygenic scores for 125 different traits representing a variety of quantitative phenotypes and diseases, and a large set of metabolites measured in up to 83,000 participants in the UK Biobank. These associations are all available via a public browser, and may be used to identify candidate intermediate phenotypes, as well as potential biomarkers of disease.

## Introduction

Complex traits and disease have a polygenic architecture meaning that they are influenced by many genetic variants scattered throughout the human genome (*Boyle et al., 2017*). An increasingly popular approach to predict disease risk in a population is to derive weighted scores by summing the number of risk increasing variants that participants harbour. These are typically referred to as 'polygenic scores' (PGS) (*Torkamani et al., 2018*; *Lewis and Vassos, 2020*). In the last decade, PGS have emerged as powerful tools for predicting lifelong risk of disease, which is predominantly due to the dramatic increase in sample sizes of genome-wide association studies (GWAS) and their continued success in uncovering trait-associated genetic variants across the genome (*Visscher et al., 2017*). Additionally, PGS have utility in a causal inference setting to establish causal effects between risk factors and disease outcomes, as well as to help elucidate putative diagnostic and prognostic biomarkers for disease incidence (*Richardson et al., 2019b*, *Holmes and Davey Smith, 2019*; *Ritchie et al., 2021a*).

The human metabolome consists of over 100,000 small molecules and is a rich source of potential risk factors and biomarkers, as well as therapeutic targets (*Holmes et al., 2021*), for complex traits and disease (*Gallois et al., 2019*). Many circulating metabolic traits studied to date have a large heritable component as demonstrated by GWAS endeavours (*Suhre et al., 2011*; *Shin et al., 2014*; *MacTel Consortium et al., 2021*), suggesting that they have a polygenic architecture. In-depth molecular profiling has recently been undertaken in the UK Biobank (UKB) study using nuclear magnetic resonance (NMR) to capture measures of 249 circulating metabolites in approximately 120,000 participants who also have genotype data (*Julkunen et al., 2021*; *Sudlow et al., 2015*). The 249 metabolite measurements include the particle concentration, size, and composition of 14 lipoprotein subclasses, as well as the levels of phospholipids, fatty acids, amino acids, ketone bodies, and other biomarkers as discussed in a recent review (*Ala-Korpela et al., 2022*). This resource therefore provides an unprecedented opportunity to characterize metabolic profiles for disease risk by leveraging genome-wide variation captured by PGS. There are multiple advantages to this approach over conventional observational associations between metabolites and complex traits or endpoints. For example, as UKB is a prospective cohort study many diseases have low prevalence, such as Alzheimer's disease which typically has a late onset. In contrast, evaluations using PGS will likely yield higher statistical power given that a continuous genetic score will be analysed for all participants in UKB based on their liability to disease.

In this study, we sought to construct an atlas of associations between 125 PGS and the 249 circulating metabolic traits in the UKB study (*Figure 1*). We demonstrate the usefulness of this atlas in terms of highlighting putative risk factors and biomarkers for disease risk and advocate the use of an approach known as Mendelian randomization (MR) to investigate whether a causal relationship may underlie findings (Supplementary Note 2) (*Davey Smith and Ebrahim, 2003*, *Davey Smith and Hemani, 2014*; *Sanderson et al., 2022*). As an exemplar, we apply MR systematically to investigate all PGS associations highlighted from a hypothesis-free scan of the inflammatory marker glycoprotein acetyls (GlycA). Furthermore, all PGS analyses were initially conducted in the full UKB sample, as well as in sex-stratified samples and age tertiles as proposed previously to evaluate the influence of medication use on findings (*Bell et al., 2022*). As the age of individuals in UKB is unlikely to induce sources of biases into analyses (e.g. collider bias), age-dependent stratification allows comparisons between the youngest and oldest tertiles in UKB where the level of medication use is likely to vary between groups. Stratification on medication use, by contrast, would introduce collider bias. Together our findings provide valuable insights into the effects of PGS on metabolic markers which may influence hypothesis generation and facilitate similar analyses to those presented in this paper.

## Results

### Constructing an atlas of polygenic score associations across the human metabolome

We obtained genome-wide summary statistics for 125 different complex traits and diseases from large-scale GWAS and constructed PGS for each of these in the UKB study. The majority of these summary statistics were obtained from the OpenGWAS platform and encompassed traits and disease outcomes from across the human phenome (*Elsworth et al., 2020*). GWAS were identified based on those conducted in populations of European descent given that our analysis in UKB was based on the

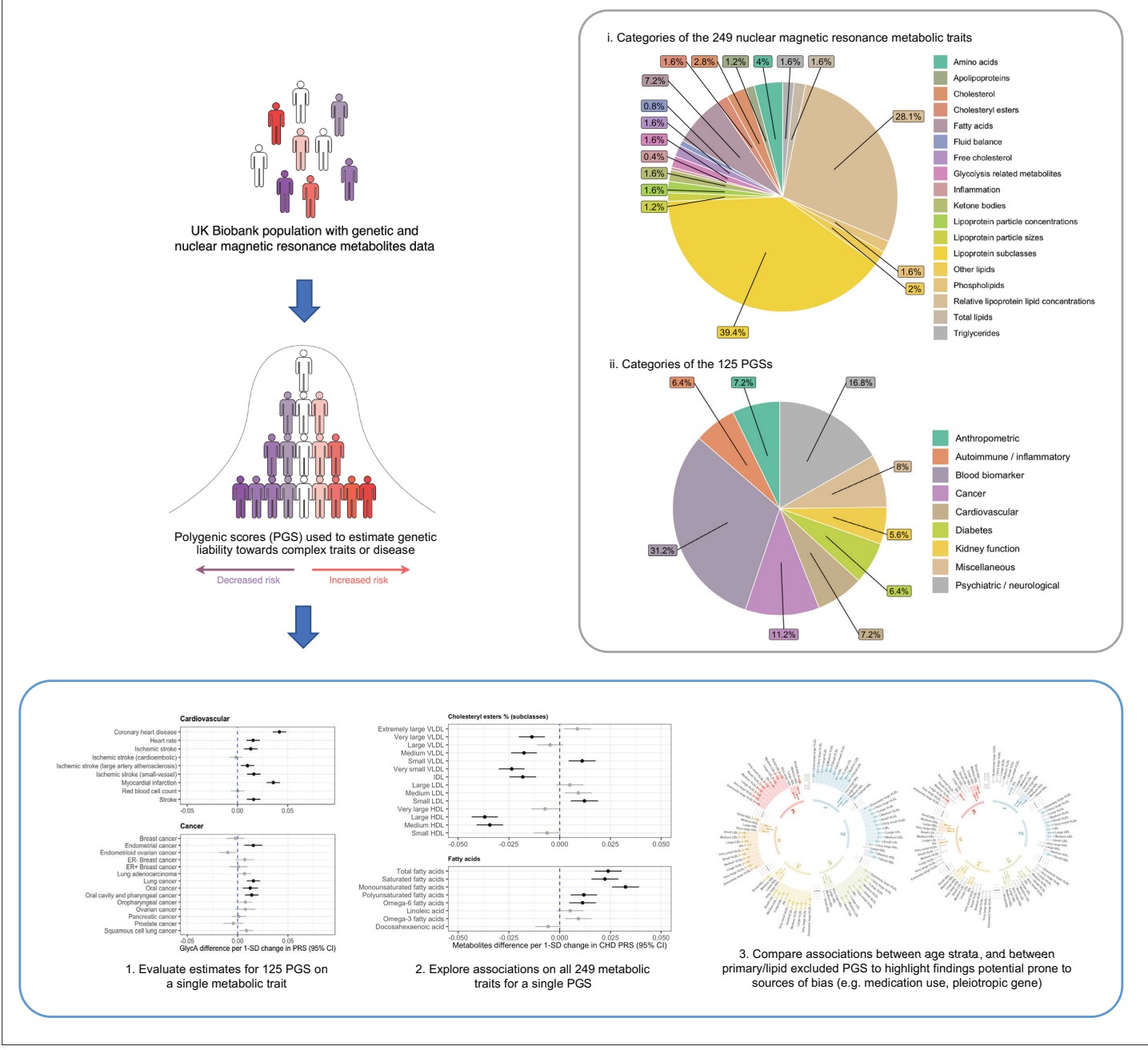

**Figure 1.** A schematic diagram depicting the data composition and analytical approach undertaken in this study.

European subset with NMR data. Furthermore, we identified studies which did not include the UKB in their study to avoid overlapping samples between PGS construction and analysis with metabolic traits. Full details for all GWAS can be found in *Supplementary file 1a*. Two versions of each PGS were built using different thresholds for variant-trait associations (P) and linkage disequilibrium (LD; $r^2$). These were (1) 'lenient' thresholds of $p < 0.05$ and $r^2 < 0.1$ and (2) a 'stringent' threshold of $p < 5 \times 10^{-8}$ and $r^2 < 0.001$. PGS were generated using the software PLINK (*Chang et al., 2015*) with LD being calculated using a reference panel of 10,000 randomly selected unrelated UKB individuals of European descent (*Kibinge et al., 2020*). The specific weights for clumped variants used in all PGS can be found at https://tinyurl.com/PRSweights.

We investigated the association between each PGS in turn with 249 circulating metabolites measured using targeted high-throughput NMR metabolomics from Nightingale Health Ltd (biomarker

quantification version 2020) (*Supplementary file 1b*; *Julkunen et al., 2021*). Our final sample size of *n* = 83,004 was determined based on individuals with both genotype and circulating metabolites data after removing participants with withdrawn consent, evidence of genetic relatedness or who were not of 'white European ancestry' based on a *K*-means clustering (*K* = 4). All PGS were standardized to have a mean of 0 and standard deviation of 1 and similarly all metabolites were subject to inverse rank normalization transformations prior to analysis allowing cross-PGS/metabolite comparisons to be made. Analyses were conducted using linear regression adjusting for age, sex, and the top 10 principal components (PCs).

To disseminate all findings from this large-scale analysis we have developed a web application (http://mrcieu.mrsoftware.org/metabolites_PRS_atlas/) to query and visualize metabolic signatures for a given PGS. In this paper, we have discussed findings using PGS that were derived using the more lenient criteria (i.e. $p < 0.05$ and $r^2 < 0.1$), although all findings based on both thresholds can be found in the web atlas. PC analysis suggested that the first 19 PCs captured 95% of the variance in the NMR metabolites data (whereas the first ten PCs captured 90% and the first 41 PCs captured 99% of the variance) (*Supplementary file 1c*). We therefore have applied a heuristic of $p < 0.05/19$ in this manuscript to account for multiple testing of the associations between any single PGS and the NMR metabolic traits for downstream analyses, although users are able to download the full results to apply whatever correction they see fit. For all other analyses (e.g. associations between metabolic traits and all PGS), we apply a false discovery rate (FDR) of less than 0.05 calculated from the Benjamini–Hochberg procedure to correct for multiple testing. Based on the FDR threshold of 0.05, there were a total of 5445 associations between PGS derived (derived based on 'lenient criteria' with $p < 0.05$ variants) in the full sample and NMR-assessed circulating metabolic traits. Heatmaps depicting the *Z* scores of all PGS–metabolic trait associations can be found in *Figure 2—figure supplements 1 and 2*. The PGS with the largest number of associations robust to multiple testing corrections was body mass index (BMI) (*n* = 217) (*Supplementary file 1d*). Our atlas also includes sex-stratified estimates for PGS weighted by GWAS undertaken in female only (such as breast cancer and age at menarche) and male only (e.g. prostate cancer) populations, as well as sex-stratified estimates in both females and males separately for all other PGS–metabolite associations. We encourage users interested in these sex-stratified estimates to interpret them with caution however, given the widespread sex-differential participation bias in UKB (*Pirastu et al., 2021*).

In this paper, we provide several examples of how results from this atlas can be used to generate hypotheses and pave the way for in-depth and careful evaluations of associations between PGS and circulating traits. Specifically, we believe our findings can facilitate a 'reverse gear Mendelian randomization' approach to disentangle whether associations likely reflect metabolic traits acting as a cause or consequence of disease risk (*Holmes and Davey Smith, 2019*) as illustrated using triglyceride-rich very-low-density lipoprotein (VLDL) particles in the next section. Furthermore, in-depth evaluations allow careful consideration of appropriate instrumental variables for circulating metabolites which can be a challenging task as highlighted in our exemplar analysis of GlycA. Finally, we provide examples of how the plethora of sensitivity analyses within our atlas can help users further investigate the robustness of findings.

## Orienting the direction of effect between putative causal relationships using Mendelian randomization

Many top associations across PGS were consistent with the known underlying biology of their corresponding diseases, as well as various proof of concepts that associations between PGS and metabolic traits may reflect both causes of disease and consequences of genetic liability towards disease. For example, we applied MR to further evaluate associations highlighted in our atlas with VLDL particles, where both VLDL particle average diameter size and concentration were associated with the PGS for BMI (Beta = 0.04, 95% CI = 0.033 to 0.046, $p = 3.53 \times 10^{-35}$ and Beta = 0.012, 95% CI = 0.006 to 0.019, $p = 2.7 \times 10^{-4}$, respectively) and also coronary heart disease (CHD) liability (Beta = 0.026, 95% CI = 0.019 to 0.032, $p = 2.12 \times 10^{-15}$ and Beta = 0.035, 95% CI = 0.028 to 0.042, $p = 2.73 \times 10^{-24}$, respectively). Conducting bidirectional MR suggested that the associations with average diameter of VLDL particles are likely attributed to a consequence of BMI and CHD liability as opposed to the size of VLDL particles having a causal influence on these outcomes (*Supplementary file 1e*). In contrast, MR analyses suggested that the concentration of VLDL particles increases risk of CHD (Beta = 1.28 per

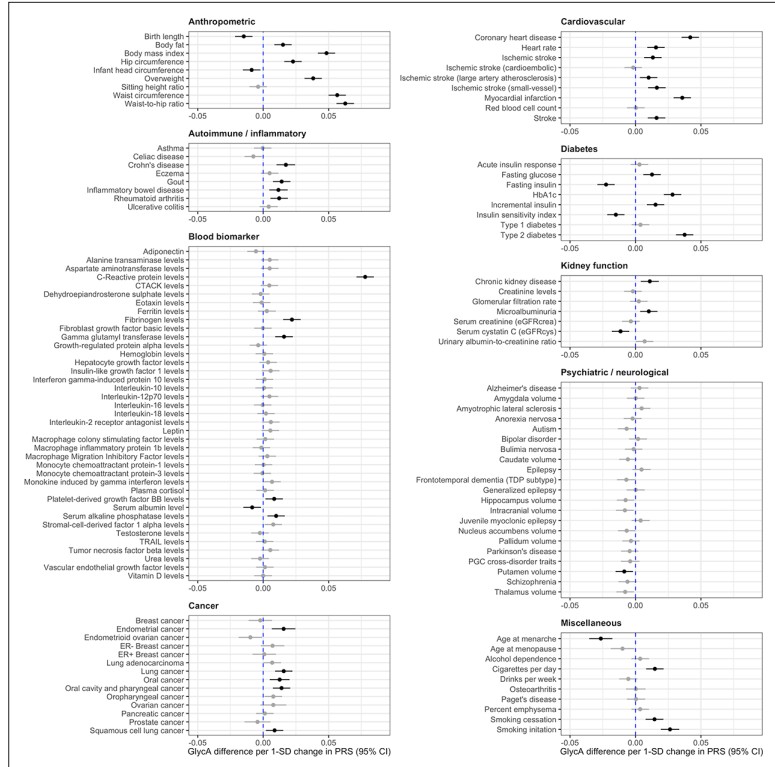

**Figure 2.** Forest plots depicting results from a systematic evaluation of 125 polygenic scores and their associations with circulating glycoprotein acetyls (GlycA). Associations were assessed by linear regression on up to 83,004 individuals in the UK Biobank. Error bars represent the 95% confidence intervals for the effect estimates. Results coloured in grey are associations which did not surpass a false discovery date of less than 0.05 to account for multiple testing.

The online version of this article includes the following figure supplement(s) for figure 2:

**Figure supplement 1.** Heatmap showing the *Z* scores of associations between metabolites and PGS that were derived using the lenient criteria (i.e. variant-trait associations with p < 0.05 and linkage disequilibrium (LD) $r^2$ < 0.1).

**Figure supplement 2.** Heatmap showing the *Z* scores of associations between metabolites and PGS that were derived using the more stringent criteria (i.e. variant-trait associations with p < 5 × 10$^{-8}$ and linkage disequilibrium (LD) $r^2$ < 0.001).

---

1-SD change in VLDL particle concentration, 95% CI = 1.25 to 1.65, p = 2.8 × 10$^{-7}$) which may explain associations between the CHD PGS and this metabolic trait within our atlas. Similar MR analyses to investigate findings from our atlas can be conducted using the full GWAS summary statistics for all 249 circulating metabolic traits available via the GWAS catalog (https://www.ebi.ac.uk/gwas/) under accession IDs GCST90092803 to GCST90093051 (*Richardson et al., 2022*).

Along with comparing metabolic signatures for a given PGS, our atlas facilitates hypothesis-free evaluations to inspect all PGS associations for a given metabolic trait. As an example of this, we have undertaken such an analysis based on the associations between all 125 PGS in our atlas with circulating GlycA. GlycA is a biomarker of chronic inflammation and has been found to predict various endpoints, including types of cardiovascular disease, cancer, and all-cause mortality (*Lawler et al., 2016*; *Connelly et al., 2017*). Although previous studies of genetically predicted GlycA have been conducted for hypotheses regarding single endpoints (*Lord et al., 2021*), whether or not circulating GlycA has a causal effect on outcomes from across the disease spectrum has yet to be comprehensively investigated. The role of GlycA is important to establish given the emerging role of inflammation as a pharmacologically modifiable pathway for the prevention and treatment of cardiovascular disease.

There were 44 PGS associations with GlycA which were robust to an FDR <5%, used as a heuristic to determine which results to investigate in further detail (*Figure 2* and *Supplementary file 1g*). We firstly applied the inverse variance weighted (IVW) MR method to systematically assess whether genetic liability to any of these disease endpoints or complex traits provided evidence of an effect on GlycA levels. Of the 44 PGS, 36 contain one or more genetic variants that reached genome-wide significance which can be used as instrumental variables for MR. In total, eight of these exposures provided evidence of a genetically predicted effect from MR analyses based on FDR <5% (*Supplementary file 1g*), which included anthropometric traits such as BMI (Beta = 0.16 SD increase in GlycA levels per 1 SD increase in BMI, 95% CI = 0.11 to 0.21, FDR = $1.59 \times 10^{-8}$) and genetic liability to cigarettes smoked per day (Beta = 0.27 SD change GlycA levels per 1 SD increase in cigarettes per day, 95% CI = 0.20 to 0.34, FDR = $2.84 \times 10^{-12}$). Estimates based on the IVW method were typically supported by the weighted median approach, although only cigarettes smoked per day were supported by both the weighed median (Beta = 0.24, 95% CI = −0.16 to 0.33, p = $2.08 \times 10^{-8}$) and MR-Egger (Beta = 0.22, 95% CI = 0.07 to 0.37, p = 0.02) methods (*Supplementary file 1h*).

Next, we investigated the converse direction of effect using MR to assess whether genetically predicted GlycA may influence any of the 44 complex traits or disease endpoints highlighted by our atlas of results. Undertaking a GWAS of GlycA in the UKB identified 59 independent genetic variants which were harnessed as instrumental variables (mean *F* = 100.1) (*Supplementary file 1i*). In contrast to the previous analysis, we identified very weak evidence using the IVW method that genetically predicted GlycA has an effect on any of the 44 traits or diseases assessed based on FDR <5% (*Supplementary file 1j*). We also conducted further sensitivity analyses given that the NMR signal of GlycA is a composite signal contributed by the glycan *N*-acetylglucosamine residues on five acute-phase proteins, including alpha1-acid glycoprotein, haptoglobin, alpha1-antitrypsin, alpha1-antichymotrypsin, and transferrin (*Otvos et al., 2015*). Using cis-acting plasma protein (where possible) and expression quantitative trait loci (pQTLs and eQTLs respectively) (*F*-stats range from 43.9 to 4468.0) as instrumental variables for these proteins (*Supplementary file 1k*) did not provide convincing evidence that they play a role in disease risk for associations between PGS and GlycA (*Supplementary file 1l*). The only effect estimate robust to multiple testing was found for higher genetically predicted alpha1-antitrypsin levels on gamma glutamyl transferase (GGT) levels (Beta = 0.05 SD change in GGT per 1 SD increase in protein levels, 95% CI = 0.03 to 0.07, FDR = $3.6 \times 10^{-3}$), although this was not replicated when using estimates of genetic associations with GGT levels from a larger GWAS conducted in the UKB data (Beta = $1.6 \times 10^{-3}$, 95% CI = $−6.9 \times 10^{-3}$ to 0.01, p = 0.71). For details of pleiotropy robust analysis and replication results see *Supplementary file 1m*.

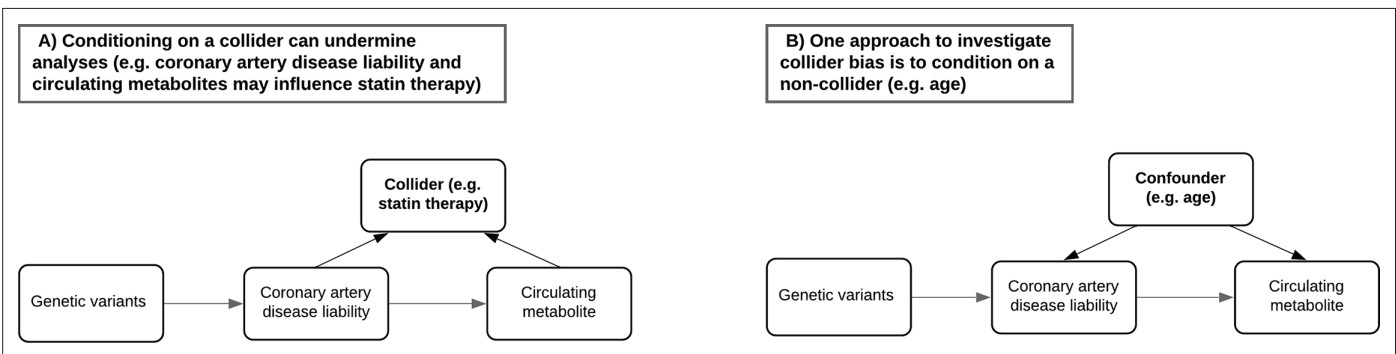

**Figure 3.** Directed acyclic graphs illustrating the potential collider bias involved in the causal relationship between the coronary artery disease polygenic score and circulating metabolites. (**A**) The likelihood of participants in UK Biobank taking medication such as statins is influenced by having a higher genetic predisposition to coronary artery disease but may also be influenced by certain metabolic traits measured on the nuclear magnetic resonance (NMR) panel (e.g. having elevated low-density lipoprotein cholesterol levels). Either stratifying or adjusting for statin use in regression models may therefore induce collider bias into the association between disease liability and metabolic traits. (**B**) Age is commonly adjusted for in association analyses due to its role as a confounder and cannot be a collider (i.e. exposures and outcomes cannot influence the age of participants). Stratifying samples by age therefore enables the analysis of exposure–outcome associations in a group of participants with relatively consistent confounding effect from age, leading to more robust association estimates in the lower age tertile where the percentage of participants who are regularly taking medication is low. Furthermore, comparisons with participants in the highest age tertile can help highlight associations between polygenic scores and metabolic traits most likely distorted by potential colliders such as statins in the full sample.

## Stratifying analyses by age to investigate potential sources of bias induced by medication use

A critical challenge when analysing the NMR metabolites data in UKB concerns the most appropriate manner to account for participants taking medications which may undermine inference (*Bell et al., 2022*). For example, UKB participants undergoing statin therapy will likely have altered levels of lipoprotein lipid metabolites compared to others. However, adjusting for statin therapy as a covariate or by stratification can induce collider bias, which may be encountered when investigating the relationship between two factors (such as genetic liability towards CHD and a lipoprotein lipid metabolite) when both influence a third factor (e.g. statin therapy) (*Figure 3A*). In particular due to the large sample sizes provided, collider bias in the UKB study has been shown to distort findings (*Griffith et al., 2020*) and in extreme cases can even result in opposite conclusions being drawn (*Richardson et al., 2019a*). Therefore, to investigate the influence of medication use on the results within our atlas, we repeated all analyses stratified by age tertiles as proposed previously (*Bell et al., 2022*), given that age is very unlikely to act as a collider between PGS and circulating metabolites (*Figure 3B*), and medication use is lower in the younger tertiles. Comparisons between the youngest and oldest tertiles in UKB can be systematically investigated and visualized using our web application to evaluate how medications may bias findings.

As an example of this, in the full UKB sample there were 193 circulating metabolites associated with the PGS for CHD (constructed using genetic variants with $p < 0.05$ and $r^2 < 0.1$) under a p value $<0.05/19$ for multiple testing correction (*Supplementary file 1n*). The vast majority of these were lipoprotein lipid traits, which are likely capturing causal risk factors for CHD. Amongst the top associations for this PGS was apolipoprotein B (apo B) (Beta = 0.027, 95% CI = 0.020 to 0.033, $p = 7.2 \times 10^{-15}$), which acts as an index of the number of circulating atherogenic lipoprotein particles and has been postulated previously to be the predominating lipoprotein lipid trait indexing CHD risk (*Ference et al., 2019*; *Sniderman et al., 2019*; *Richardson et al., 2020b, Richardson et al., 2021*).

Evaluating this association between age tertiles allowed us to investigate whether it may be influenced by medications in UKB, such as the impact of statin therapy on lowering low-density lipoprotein (LDL) cholesterol, which apo B particles carry. In the youngest tertile (mean age = 47.3 years, 5%

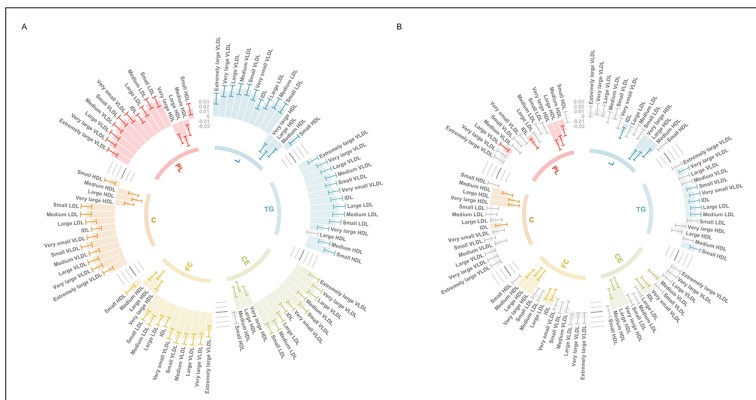

**Figure 4.** Circos plots illustrating the utility of age-stratified analyses in UK Biobank to investigate potential sources of bias when evaluating associations between polygenic scores (PGS) and circulating metabolites. Associations were assessed by linear regression on up to 83,004 individuals in the UK Biobank. Error bars represent confidence intervals of the effect estimates between the coronary heart disease (CHD) PGS and traits from the six subclasses: L = total lipids, TG = triglycerides, CE = cholesteryl esters, FC = free cholesterol, C = cholesterol, PL = phospholipids. Grey bars represent associations not robust to multiple testing based on $p > 0.05/19$. These barcharts are oriented such that those extending to the outer rim reflect a positive association between the CHD PGS and metabolic traits whereas those extending inwards indicate inverse associations. (**A**) Analyses undertaken for participants in the lowest age tertile (mean age = 47.3 years, 5% statin users) and (**B**) the corresponding results for the oldest age tertile (mean age = 65.3 years, 29% statin users).

The online version of this article includes the following figure supplement(s) for figure 4:

**Figure supplement 1.** A forest plot illustrating the associations between the polygenic score (PGS) for coronary heart disease (CHD) with 249 circulating metabolic traits in UK Biobank.

statin users), the association between the CHD PGS with apo B was markedly stronger than in the total sample (Beta = 0.059, 95% CI = 0.048 to 0.070, p < 1.6 × 10$^{-26}$). In stark contrast, there was weak evidence of an inverse association between apo B and the CHD PGS in the oldest tertile (mean age = 65.3 years, 29% statin users) (Beta = −0.007, 95% CI = −0.019 to 0.004, p = 0.223), which is likely attributed to the higher proportion of participants undergoing statin therapy in this sample. Similarly, concentrations of VLDL, LDL, and IDL provided evidence of a positive association with the CHD PGS in the youngest tertile (*Figure 4A*), whereas the corresponding associations in the oldest tertile provided weak evidence of association (and in some cases reversed direction entirely) (*Figure 4*). A comparison of all 249 associations with the CHD PGS derived in the youngest and oldest age tertiles can be found in *Supplementary file 1o*.

### Elucidating polygenic associations with metabolic traits by excluding major regulators of NMR lipoprotein lipids

The polygenic nature of complex traits means that the inclusion of highly weighted pleiotropic genetic variants in PGS may introduce bias into genetic associations within our atlas. To provide insight into this issue, we constructed PGS excluding variants within the regions of the genome which encode the genes for 13 major regulators of NMR lipoprotein lipids signals which captured 75% of the gene–metabolite associations in the Finnish Metabolic Syndrome In Men (METSIM) cohort (*Gallois et al., 2019*). For details of these genes see *Supplementary file 1p*.

For PGS with these lipid loci excluded, anthropometric traits such as waist-to-hip ratio (*N* = 209), waist circumference (*N* = 206), and BMI (*N* = 205) still provided strong evidence of association with the majority of metabolic measurements on the NMR panel based on multiple testing corrections. Elsewhere however, the Alzheimer's disease PGS, which was associated with 60 metabolic traits robust to p < 0.05/19 in the initial analysis including these lipid loci (*Supplementary file 1q*), provided no convincing evidence of association with the 249 circulating metabolites after excluding the lipid loci based on the same multiple testing threshold (*Supplementary file 1r*). Further inspection suggested that the likely explanation for this attenuation of evidence were due to variants located within the *APOE* locus (near one of the major lipid regulators *APOC1*) which are recognized to exert their influence on phenotypic traits via horizontally pleiotropic pathways (*Ferguson et al., 2020*).

## Discussion

In this study, we have developed an atlas of polygenic risk score associations with circulating metabolic traits in an unprecedented sample size compared to previous studies. Our results can be used to help prioritize findings worthy of follow-up, using techniques such as MR, as a means of disentangling putative causal and non-causal relationships underlying associations between PGS and circulating biomarkers. Furthermore, conducting all analyses within age tertiles illustrates the potential of medication use within the UKB population to bias relationships within our atlas of results. These results should help highlight disease–metabolic trait relationships where researchers should exercise caution when interpreting findings from their own analyses of the recently generated NMR metabolites data in UKB, which are due to be available in all ~500,000 participants in the forthcoming years.

Amongst the thousands of PGS associations identified in this study with a p value <0.05/19, we observed an enrichment of scores derived using GWAS of anthropometric traits. This was exemplified by the waist circumference PGS which yielded the largest number of associations in our atlas (*n* = 212). Previous studies in the field have demonstrated the strong influence that adiposity has on circulating traits from across the metabolome (*Würtz et al., 2014*), and indeed across the proteome (*Folkersen et al., 2020*). Furthermore, as shown previously by an MR study (*Bell et al., 2022*), certain associations with the BMI PGS may be due to the influence of medication use in the UKB sample, for example those related to LDL (e.g. BMI PGS on total lipids in LDL: Beta = −0.020, 95% CI = 0.027 to −0.013, p < 6.17 × 10$^{-9}$). In our atlas, evidence of these associations strongly attenuated in the youngest age tertile, where the influence of such factors in the UKB population may be weakest (e.g. total lipids in LDL from youngest age tertile: Beta = 0.005, 95% CI = −0.006 to 0.016, p = 0.35). In addition to the striking difference highlighted in our study between the CHD PGS and apolipoprotein B across age tertiles, findings such as this further emphasize the importance of evaluating results from the full sample analysis together with those derived in age-stratified subsamples. Moreover, we suggest that

users interested in the sex-stratified estimates within our atlas should interpret them in conjunction with estimates derived across age tertiles as in this example, given that the proportions of males and females in UKB taking certain medications may differ (e.g. statins).

As an exemplar, we conducted a hypothesis-free evaluation of one of the metabolic traits on the UKB NMR panel, GlycA, as a means of demonstrating how findings from our atlas may help generate hypotheses and follow-up analyses. Whilst our MR results indicated that modifiable risk factors such as BMI and cigarette smoking may increase levels of this circulating inflammatory biomarker, they suggest that targeting GlycA itself is unlikely to yield a beneficial therapeutic effect on the complex traits and disease endpoints evaluated in this study. This highlights the value of findings from our atlas, complemented by approaches such as MR, to help both prioritize and deprioritize circulating metabolic traits for further evaluation. Similar hypothesis-free evaluations on the other 248 metabolic traits can be routinely undertaken using our web tool, in addition to evaluations using the more stringent PGS construction criteria of $p < 5 \times 10^{-8}$ and $r^2 < 0.001$. We reiterate the importance of using approaches such as MR (including sensitivity analyses, which are at least partially robust to various forms of pleiotropy) to formally assess putative causal relationships which may underlie findings in our atlas however, as well as to help orient their directionality. This is particularly important given that PGS may be more prone to recapitulating sources of bias commonly encountered in observational studies in comparison to formal MR analyses (*Richardson et al., 2019b*, *Ritchie et al., 2021a*). We likewise conducted bidirectional MR to demonstrate that associations between the CHD PGS and VLDL particle size likely reflect an effect of CHD liability on this metabolic trait. In contrast, the association between the CHD PGS and VLDL concentrations are likely attributed to the causal influence of this metabolic trait on CHD risk, suggesting that it is the concentration of these triglyceride-rich particles that are important in terms of the aetiology of CHD risk as opposed to their actual size. We believe that findings from our atlas, as well as other ongoing efforts which leverage the large-scale NMR data within UKB, should facilitate further granular insight into lipoprotein lipid biology.

In terms of study limitations, we note that the NMR panel is predominantly focussed on lipoprotein lipids and as such our atlas does not facilitate analyses across the entire metabolome. Availability of metabolomics quantified by other platforms (e.g. mass spectroscopy) in large numbers with GWAS genotyping will aid in this effort (*MacTel Consortium et al., 2021*). Furthermore, whilst these data provide an unparalleled sample size compared to predecessors, findings are based on traits derived from whole blood and may therefore not be reflective of molecular signatures identified in other tissue types (*Richardson et al., 2020a*). In terms of interpretation, we emphasize that PGS can capture an estimate of an individual's lifelong disease risk, and as such results based on the UKB NMR metabolites dataset, measured at a midlife timepoint in the lifecourse in predominantly healthy participants, may differ substantially to metabolomic profiles of patients with a disease. Conversely, findings may hold the potential to highlight biomarkers useful for disease prediction before clinical manifestation, therefore indicating a potential window of opportunity for early detection and/or intervention. Lastly, in this study we leveraged data from the European subset of the UKB study, which may therefore not be representative of individuals from other ancestries (*Duncan et al., 2019*). Larger sample sizes of non-European individuals with metabolomics data will facilitate analyses in other ancestries once available, in addition to findings from future large-scale GWAS which have been principally confined to individuals of European descent to date (*Sirugo et al., 2019*).

We envisage that findings from our atlas will motivate future study hypotheses and help prioritize (and deprioritize) circulating metabolic traits for further in-depth research. Although we highlight several key findings in this manuscript, all our findings can be queried using our web application which provides a platform to inform researchers in the field planning similar analyses. Similar evaluations to those conducted in this manuscript should help develop a deeper understanding into how circulating metabolic traits contribute towards complex trait variation and assess their putative mediatory roles along the causal pathways between modifiable lifestyle risk factors and disease endpoints.

## Methods

### Data sources

#### The UKB study

Metabolic profiling was undertaken on a random subset of individuals from the UKB study (*Sudlow et al., 2015*) (range between 116,353 and 121,695). Full details on genotyping quality control (QC), phasing and imputation in UKB have been described previously (*Bycroft et al., 2018*). In brief, samples were restricted to individuals of white British ancestry who self-report as 'White British' and who have very similar ancestral backgrounds according to PC analyses performed by Bycroft et al (*n* = 409,703). In total, 107,162 pairs of related individuals were removed based on estimated kingship coefficients derived using the KING toolset. An in-house algorithm was then applied to preferentially remove individuals related to the greatest number of other individuals until no related pairs were left (removing *n* = 79,448 in total). A further 2 individuals were removed as they were related to over 200 other individuals. There were *n* = 814 individuals with sex-mismatch (derived by comparing genetic sex and reported sex) or individuals with sex chromosome aneuploidy excluded from analyses.

In total, 249 metabolic biomarkers were generated using non-fasting plasma samples (aliquot 3) taken from UKB participants at initial or subsequent clinical visits. Targeted high-throughput NMR metabolomics from Nightingale Health Ltd (biomarker quantification version 2020) were used to generate data on each of the 249 measures. These included biomarkers on lipoprotein lipid traits, their concentrations and subclasses, fatty acids, ketone bodies, glycolysis metabolites, and amino acids. Further details are described elsewhere (*Julkunen et al., 2021*). For QC, data of the 249 NMR metabolomics traits were processed using the 'ukbnmr' R package to remove variation due to technical factors caused by differences in sample handing and measurement (*Ritchie et al., 2021b*). Statin users in UKB were identified based on medication codes as defined previously (*Sinnott-Armstrong et al., 2021*). A full list of these metabolic biomarkers and their summary characteristics can be found in *Supplementary file 1b*.

Ethical approval for this study was obtained from the Research Ethics Committee (REC; approval number: 11/NW/0382) and informed consent was collected from all participants enrolled in UKB. Data were accessed under UKB application #15825 and #81499.

#### GWAS summary statistics

Publicly available GWAS summary statistics were extracted from the OpenGWAS platform (https://gwas.mrcieu.ac.uk/) and publicly available repositories (*Elsworth et al., 2020*). We identified GWAS for 125 different complex traits and diseases which were selected to encompass a broad range of human phenotypes for which genome-wide data were available allowing us to construct PGS based on all variants available with p < 0.05. Furthermore, we identified GWAS based on study populations with participants of European descent, as our study was based on the unrelated European participants of UKB with NMR measures, as well as studies which had not analysed the UKB study population to avoid overlapping samples which can lead to overfitting bias in results (*Fang et al., 2022*). All details of these GWAS can be found in *Supplementary file 1a*.

### PGS construction

We built two versions of each PGS in this study using the following criteria. Firstly, scores were developed with independent variants (i.e. $r^2 < 0.001$) which were robustly associated with their traits or disease based on conventional genome-wide corrections (i.e. $p < 5 \times 10^{-8}$). The second versions of scores were derived using more lenient thresholds which were $r^2 < 0.1$ and p < 0.05. LD to estimate correlation between variants was based on a previously constructed reference panel of 10,000 randomly selected unrelated UKB individuals of European descent (*Kibinge et al., 2020*). PGS were derived for all participants with both genotype and NMR metabolites data after firstly excluding individuals with withdrawn consent, evidence of genetic relatedness or who were not of 'white European ancestry' based on a *K*-means clustering (*K* = 4). These scores were built by summing trait/disease risk increasing alleles which participants harboured weighted by their effect size reported by GWAS using genotype data from hard call dosages files (plink binaries bed/bim/fam) and the software PLINK v2.0 (*Chang et al., 2015*).

The majority of PGS were constructed in all eligible participants, with the exception of those based on GWAS in sex-stratified populations. These were breast cancer (including ER+ and ER− PGS), endometrial cancer, ovarian cancer, endometrioid ovarian cancer, age at menarche, age at menopause, and bulimia nervosa, which were derived in females only, as well as the prostate cancer PGS derived in males only. Additionally, we also built two versions of PGS for all complex traits excluding variants at 13 lipid-associated gene loci, which were *DOCK7*, *CELSR2*, *GALNT2*, *PCSK9*, *GCKR*, *TRIB1*, *LPL*, *APOA5*, *FADS2*, *LIPC*, *CETP*, *LDLR*, and *APOC1* (consisting of the encoding gene region itself as well as a 1 Mb window either side). Details of the 13 genes are presented in ***Supplementary file 1p***.

## Statistical analysis

### PC analysis of the metabolites

PC analysis was performed on the post-QC metabolite data to identify the number of independent traits among the 249 highly correlated metabolites. The analysis was conducted using the *princomp* function from the 'stats' R package.

### PGS analysis

To allow us to draw comparisons between PGS–metabolite associations, we standardized all PGS to have a mean of 0 and standard deviation of 1 and additionally applied inverse rank normalization transformations to all metabolic traits prior to analysis. Associations between PGS and normalized metabolites were determined by linear regression with adjustment for age, sex (where appropriate), genotyping chip, the top 10 PCs, and fasting time. Each analysis was conducted initially in the full sample, followed by analyses after stratification into age tertiles to investigate the influence of medication use on findings. Sex-stratified association analyses in the full sample were also conducted whereby metabolic traits were transformed separately among males and females before applying linear regression. All analyses were undertaken using both versions of each PGS as long as their corresponding GWAS had at least one variant with $p < 5 \times 10^{-8}$ necessary for the more stringent criteria. To account for multiple testing in this study, we applied a p value threshold of 0.05/19 (accounting for 19 independent variables captured 95% variances in the metabolites from PC analysis) to highlight findings worthwhile evaluating in further detail. However, all results from our analyses are available in the web application should users decide to apply a more stringent (or lenient) heuristic to prioritize findings for in-depth analyses.

### Instrument selection for GlycA analysis

Genetic instruments for all PGS traits/disease points evaluated in this study using MR were obtained from the 'TwoSampleMR' v0.5.6 R package (***Hemani et al., 2018***), or by manually uploading GWAS summary statistics and using the *clump_data* function from this package to identify them. Instruments for GlycA and particle size of very low-density lipoprotein (VLDL) were identified by conducting a GWAS of this trait in the UKB study using the BOLT-LMM (linear mixed model) software to control for population structure (***Loh et al., 2015***). Analyses were undertaken after excluding individuals of non-European descent (based on *K*-mean clustering of *K* = 4) and standard exclusions, including withdrawn consent, mismatch between genetic and reported sex, and putative sex chromosome aneuploidy. Analyses were adjusted for age, sex, fasting status, and a binary variable denoting the genotyping chip used in individuals (the UKBB Axiom array or the UK BiLEVE array). Genetic instruments were defined as variants with $p < 5 \times 10^{-8}$ after removing those in LD using the *clump_data* function as above. The full GWAS summary statistics for GlycA as well as the other 248 circulating metabolic traits are available via the GWAS catalog (https://www.ebi.ac.uk/gwas/) under accession IDs GCST90092803 to GCST90093051 (***Richardson et al., 2022***).

The NMR signal of serum GlycA is contributed by five acute-phase proteins (alpha1-acid glycoprotein, haptoglobin, alpha1-antitrypsin, alpha1-antichymotrypsin, and transferrin) (***Otvos et al., 2015***). Thus, another set of genetic instruments for GlycA were selected among variants strongly associated with these five proteins for further evaluation of the genetically predicted effect of GlycA. Instrumental variables for alpha1-acid glycoprotein were identified usinge eQTLs ($p < 5 \times 10^{-8}$, $r^2 < 0.001$) for *ORM1* from the eQTLGen Consortium (***Võsa et al., 2021***) due to a lack of available protein data. Genetic instruments for haptoglobin, alpha1-antitrypsin, alpha1-antichymotrypsin, and transferrin were identified using pQTLs ($p < 5 \times 10^{-8}$, $r^2 < 0.001$) for these proteins identified from 35,559 Icelanders

(*Ferkingstad et al., 2021*). To identify cis-acting instruments for these five proteins, we restricted e/pQTL associations to variants located within 1 Mb around their encoding genes: *ORM1* (encoding alpha1-acid glycoprotein; Ensembl ID: ENSG00000229314), *HP* (encoding haptoglobin; Ensembl ID: ENSG00000257017), *SERPINA1* (encoding alpha1-antitrypsin; Ensembl ID: ENSG00000197249), *SERPINA3* (encoding alpha1-antichymotrypsin; Ensembl ID: ENSG00000196136), or *TF* (encoding transferrin; Ensembl ID: ENSG00000196136). Independent instruments were identified using the same protocol as above (*Kibinge et al., 2020*).

## MR analyses

MR analyses were undertaken using the 'TwoSampleMR' package (*Hemani et al., 2018*) to estimate the bidirectional effects between PGS traits/disease endpoints and metabolic traits, including GlycA and VLDL particle size. This was firstly estimated using the IVW method, which takes the SNP-outcome estimates and regresses them on those for the SNP-exposure associations (*Burgess et al., 2013*), followed by the weighted median and MR-Egger methods which are considered to be more robust to horizontal pleiotropy than the IVW approach (*Bowden et al., 2015*; *Bowden et al., 2016*). If only one SNP is available as genetic instrument, Wald ratio estimates were calculated by dividing the SNP-outcome estimates by the SNP-exposure estimates (*Burgess et al., 2017*). $F$-Statistics were calculated to assess weak instrument bias (*Burgess and Thompson, 2013*). Benjamini–Hochberg FDR threshold of less than 5% was applied as a heuristic to account for multiple testing in the results.

Forest and circos plots in this study were generated using the R package 'ggplot' v3.3.3 (*Ginestet, 2011*). Heatmaps were generated using the R package 'pheatmap' v1.0.12 (*Kolde, 2015*). The web application was developed using the R package 'shiny' v1.0.4.2 (*Chang et al., 2020*). All analyses were undertaken using R (version 3.5.1).

# Acknowledgements

We would like to thank all the consortia for making their summary statistics available for the benefit of this work. We also thank the participants of the UK Biobank study. Data were accessed under UKB application #15825 and #81499. All authors work at the MRC Integrative Epidemiology Unit at the University of Bristol (MC_UU_00011/1, MC_UU_00011/4). TRG and GDS conduct research at the NIHR Biomedical Research Centre at the University Hospitals Bristol NHS Foundation Trust and the University of Bristol. The views expressed in this publication are those of the author(s) and not necessarily those of the NHS, the National Institute for Health Research, or the Department of Health. SF is supported by a Wellcome Trust PhD studentship in Molecular, Genetic and Lifecourse Epidemiology [108902/Z/15/Z]. MVH works in a unit that receives funding from the UK Medical Research Council and is supported by a British Heart Foundation Intermediate Clinical Research Fellowship (FS/18/23/33512) and the National Institute for Health Research Oxford Biomedical Research Centre.

# Additional information

### Competing interests

Michael V Holmes: MVH has consulted for Boehringer Ingelheim, and in adherence to the University of Oxford's Clinical Trial Service Unit & Epidemiological Studies Unit (CSTU) staff policy, did not accept personal honoraria or other payments from pharmaceutical companies. Tom R Gaunt: TRG receives funding from Biogen for unrelated research. Tom G Richardson: TGR is employed part-time by Novo Nordisk outside of this work. The other authors declare that no competing interests exist.

### Funding

| Funder | Grant reference number | Author |
| --- | --- | --- |
| Medical Research Council | MC_UU_00011/1 | George Davey Smith |
| Medical Research Council | MC_UU_00011/4 | Tom R Gaunt |
| British Heart Foundation | FS/18/23/33512 | Michael V Holmes |

| Funder | Grant reference number | Author |
|---|---|---|
| Wellcome Trust | 108902/Z/15/Z | Si Fang |

The funders had no role in study design, data collection, and interpretation, or the decision to submit the work for publication. For the purpose of Open Access, the authors have applied a CC BY public copyright license to any Author Accepted Manuscript version arising from this submission.

## Author contributions

Si Fang, Formal analysis, Investigation, Visualization, Methodology, Writing – review and editing; Michael V Holmes, Tom R Gaunt, George Davey Smith, Methodology, Writing – review and editing; Tom G Richardson, Conceptualization, Resources, Data curation, Formal analysis, Supervision, Funding acquisition, Validation, Investigation, Visualization, Methodology, Writing – original draft, Writing – review and editing

## Author ORCIDs

Si Fang http://orcid.org/0000-0003-4934-1212
Michael V Holmes http://orcid.org/0000-0001-6617-0879
Tom R Gaunt http://orcid.org/0000-0003-0924-3247
George Davey Smith http://orcid.org/0000-0002-1407-8314
Tom G Richardson http://orcid.org/0000-0002-7918-2040

## Ethics

Our study involves previously collected data (genomic sequencing data and metabolites data) of human participants in the UK Biobank (UKB) cohort study. Ethical approval for the UKB was obtained from the Research Ethics Committee (REC; approval number: 11/NW/0382) and informed consent was collected from all participants enrolled in UKB.

## Decision letter and Author response

Decision letter https://doi.org/10.7554/eLife.73951.sa1
Author response https://doi.org/10.7554/eLife.73951.sa2

# Additional files

## Supplementary files

• Supplementary file 1. Supplementary tables. (a) Genome-wide association studies used to derive weights for polygenic risk scores in this study. (b) Metabolic traits analyses in this study from the UK Biobank nuclear magnetic resonance (NMR) panel. (c) Principal component analysis of the NMR metabolites data. (d) Associations with the body mass index polygenic risk score. (e) Mendelian randomization results with very low-density lipoprotein (VLDL) particle size as the exposure and complex traits as the outcome. (f) Polygenic risk scores association for 125 complex traits with glycoprotein acetyls levels. (g) Mendelian randomization results for complex traits and disease liability with glycoprotein acetyls as an outcome. (h) Mendelian randomization results using weighted median and MR-Egger for complex traits and disease liability with glycoprotein acetyls as an outcome. (i) Genetic instruments for glycoprotein acetyls. (j) Mendelian randomization results with glycoprotein acetyls as our exposure and complex traits/diseases as our outcome. (k) Genetic instruments for five proteins contributing to NMR signal of glycoprotein acetyls. (l) Mendelian randomization results with each of the five acute-phase proteins as the exposure and complex traits/diseases as the outcome. (m) Mendelian randomization results using alpha1-antitrypsin as the exposure and gamma glutamyl transferase as the outcome. (n) Associations with the coronary artery disease polygenic score in the full sample. (o) Associations with the coronary artery disease polygenic risk score in the youngest and oldest age tertiles. (p) List of lipid-associated genes removed from polygenic scores in sensitivity analysis. (q) Associations with the Alzheimer's disease polygenic risk score. (r) Associations with the Alzheimer's disease polygenic risk score (excluding lipid loci).

• Transparent reporting form

**Data availability**

All data generated in this study can be downloaded from the web application of our metabolites-PGS atlas: http://mrcieu.mrsoftware.org/metabolites_PRS_atlas/.

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
