## [Editor Report]

The authors describe their work on an atlas of associations between polygenic scores for 125 different traits representing a variety of quantitative phenotypes and diseases, and a large set of metabolites measured in up to 83,000 participants in the UK Biobank. These associations are all available via a public browser, and may be used to identify candidate intermediate phenotypes, as well as potential biomarkers of disease.

---

## [Decision Letter]

**Decision letter after peer review:**

Thank you for submitting your article "Constructing an atlas of associations between polygenic risk scores from across the human phenome and circulating metabolic biomarkers" for consideration by *eLife*. Your article has been reviewed by 3 peer reviewers, one of whom is a member of our Board of Reviewing Editors, and the evaluation has been overseen by Carlos Isales as the Senior Editor. The following individuals involved in review of your submission have agreed to reveal their identity: Scott Ritchie (Reviewer #2); Maik Pietzner (Reviewer #3).

Essential revisions:

1) While this work provides a potentially important resource for the community, the examples provided tend to be illustrative, and do not really do justice to the resource. We strongly encourage the authors to revisit the data analysis for more-compelling examples of the types of impactful work that can be achieved using this resource. How and why should users want to use this Atlas?

2) One key question is whether these analyses bring us closer to causality compared to 'classical' observational associations, given LD confounding and strong metabolite variants included in PRS driving the associations. For example, how robust are PRS associations to the exclusion of individual regions? Second, can we determine whether the associations are causally upstream, or downstream of the identified phenotypes? At minimum the authors should demonstrate for examples how these issues can be teased apart (and optionally provide analyses atlas-wide).

3) The atlas would also be potentially strengthened by including additional results for sex-stratified models. Previous work has shown significant differences in NMR biomarker concentrations between males and females, so it would be valuable to see if these result in any differences in PRS associations which may arise despite PRSs largely being constructed from autosomal GWAS.

Furthermore, the reviewers provide numerous additional specific and insightful comments; we strongly encourage the authors to consider these points seriously in their revisions.

*Reviewer #1 (Recommendations for the authors):*

Specific comments:

Some of the text on figures is very small and it may be worth revisiting figure design.

P8: the positive control of creatinine and kidney disease seems a bit too trivial, given that creatinine is a diagnostic criterion. Perhaps there may be a better example?

P7 Para1: It would be helpful to the reader to say a bit more about the range of types of traits considered here, as well as typical sample sizes given that the data come from a variety of sources.

Consider using PGS instead of PRS as more accurate for quantitative and non-disease traits.

*Reviewer #2 (Recommendations for the authors):*

Introduction/discussion: this is not the first study that has demonstrated the utility of PRS associations for prioritising molecular traits for follow-up. We have recently published a paper doing so for cardiometabolic PRS and protein levels (Ritchie et al., 2021a) and this has been available as a preprint in various forms since 2019. This should be noted and cited appropriately at the relevant points in the introduction and discussion.

The associations in the atlas for the systolic blood pressure (SBP) and diastolic blood pressure (DBP) are invalid. As the authors note in the methods, PRSs for these two traits are constructed from GWAS that include UK Biobank samples. PRSs are invalid when used in samples that contributed to their underlying GWAS (Lambert et al., 2021; Wand et al., 2021; Wray et al., 2013) as this causes them to have dramatically inflated associations. Associations for these two PRS should be removed from the atlas and study, or if the authors think these two traits are of critical interest, an alternate source of GWAS that do not include UK Biobank participants should be used to construct these two PRSs.

We have found significant sources of technical variation exist in the NMR metabolomics data for UK Biobank (Ritchie et al., 2021b), which may confound some of the PRS associations here. Of particular concern in the context of this study are the presence of outlier shipping plates, on which samples have systematically high (or low) concentrations of non-biological origin (see Figure 5 in Ritchie et al. 2021b). This affects up to 5% of samples depending on the biomarker, but will have an outsized effect on associations as they result in samples being spuriously located at the extremities of biomarker distributions. I.e. this may result in associations being weaker or not significant, or less likely, false positives may be introduced if outlier plates happen to correlate with PRS. There are also a small number of biomarkers significantly impacted by other sources of technical variation, e.g. drift over time within spectrometer, or sample degradation due to extended time between sample preparation and sample measurement. We have made available an R package: ukbnmr (https://github.com/sritchie73/ukbnmr), for correcting for this technical variation, and removing samples on these outlier plates. Although this preprint is still under review, we would suggest using this package to remove the described technical variation, or at least checking whether doing so significantly changes associations in the atlas.

There are also major biological determinants of NMR biomarker concentrations that have not been accounted for, and thus may confound, PRS to biomarker associations. In addition to age, sex, and 10 genetic principal components that the authors already adjust for, NMR biomarker concentrations are also strongly correlated with body mass index (BMI), fasting time, and lipid lowering medication (statin) usage (see Figure 8 in Ritchie et al., 2021b). The atlas would be greatly improved by either appropriately accounting for these in the basic models, or including additional results using models that do so.

Regarding statin usage in particular, we appreciate that the authors have conducted a separate age-stratified analysis to evaluate the impact of medication usage on PRS to biomarker associations. However, the fact remains that statin usage will be confounding the PRS to biomarker associations in the main results (e.g. by artificially lowering LDL cholesterol in people at high CHD PRS). This confounding is typically removed in one of two ways: either (1) fitting associations excluding participants taking statins, or (2) applying biomarker-specific correction factors to concentrations prior to association analysis, such as those estimated by (Kofink et al., 2017; Sliz et al., 2018). The latter is probably the better approach as the former is likely to introduce further bias towards young and healthy participants.

The atlas would also be potentially strengthened by including additional results for sex-stratified models. We have observed significant differences in NMR biomarker concentrations between males and females, so it would be interesting to see if these result in any differences in PRS associations which may arise despite PRSs largely being constructed from autosomal GWAS.

Regarding the bi-directional Mendelian randomization analysis of GlycA, we would suggest using an alternative biomarker as exemplar in the study, as GlycA is heterogeneous and needs to be more carefully instrumented than has been done in the study currently. GlycA is an NMR signal that quantifies the total concentrations of five proteins in circulation (Otvos et al., 2015): α-1-acid glycoprotein (AGP), α-1-antitrypsin (AAT), α-1-antichymotrypsin (AACT), haptoglobin (HP), and transferrin (TF). These are acute-phase reactants whose concentrations each change in response to acute inflammation or in chronic inflammation (Connelly et al., 2017). Moreover these changes are differential with respect to each other, and over time (Ebersole and Cappelli, 2000; Gabay and Kushner, 1999). Further, a single GlycA measurement cannot be simply decomposed as two people with the same GlycA concentration can have different concentrations of the underlying proteins (Ritchie et al., 2019). This heterogeneity means that using genome-wide significant QTLs for GlycA is unlikely to be appropriate as these QTLs may include (1) protein-QTLs for one or more of the five proteins GlycA captures, and (2) signals involved in initiation of acute-phase response (e.g. interleukin-6 signalling pathways). To use GlycA as an exposure in Mendelian randomization analysis, the QTLs selected as instruments should be restricted to cis-pQTLs for the five proteins. These signals may need to be treated separately, as in the scenario GlycA is truly causal, it is unlikely that all five proteins are causal, or that they have similar causal effect sizes.

Data availability: the PRS should also be made publicly available, e.g. downloadable via the atlas. These should include at minimum: the set of variants in each PRS, variant identifier information (e.g. chromosome and position), genome build, effect allele, and weight (β or log odds from the underlying GWAS).

References

Bretherick AD, Canela-Xandri O, Joshi PK, Clark DW, Rawlik K, Boutin TS, Zeng Y, Amador C, Navarro P, Rudan I, Wright AF, Campbell H, Vitart V, Hayward C, Wilson JF, Tenesa A, Ponting CP, Baillie JK, Haley C. 2020. Linking protein to phenotype with Mendelian Randomization detects 38 proteins with causal roles in human diseases and traits. PLoS Genet 16:e1008785.

Bycroft C, Freeman C, Petkova D, Band G, Elliott LT, Sharp K, Motyer A, Vukcevic D, Delaneau O, O'Connell J, Cortes A, Welsh S, Young A, Effingham M, McVean G, Leslie S, Allen N, Donnelly P, Marchini J. 2018. The UK Biobank resource with deep phenotyping and genomic data. Nature 562:203-209.

Connelly MA, Otvos JD, Shalaurova I, Playford MP, Mehta NN. 2017. GlycA, a novel biomarker of systemic inflammation and cardiovascular disease risk. J Transl Med 15:219.

Ebersole JL, Cappelli D. 2000. Acute-phase reactants in infections and inflammatory diseases. Periodontol 2000 23:19-49.

Gabay C, Kushner I. 1999. Acute-phase proteins and other systemic responses to inflammation. N Engl J Med 340:448-454.

Kofink D, Eppinga RN, van Gilst WH, Bakker SJL, Dullaart RPF, van der Harst P, Asselbergs FW. 2017. Statin Effects on Metabolic Profiles: Data From the PREVEND IT (Prevention of Renal and Vascular End-stage Disease Intervention Trial). Circ Cardiovasc Genet 10. doi:10.1161/CIRCGENETICS.117.001759

Lambert SA, Gil L, Jupp S, Ritchie SC, Xu Y, Buniello A, McMahon A, Abraham G, Chapman M, Parkinson H, Danesh J, MacArthur JAL, Inouye M. 2021. The Polygenic Score Catalog as an open database for reproducibility and systematic evaluation. Nat Genet 53:420-425.

Otvos JD, Shalaurova I, Wolak-Dinsmore J, Connelly MA, Mackey RH, Stein JH, Tracy RP. 2015. GlycA: A Composite Nuclear Magnetic Resonance Biomarker of Systemic Inflammation. Clin Chem 61:714-723.

Ritchie SC, Kettunen J, Brozynska M, Nath AP, Havulinna AS, Männistö S, Perola M, Salomaa V, Ala-Korpela M, Abraham G, Würtz P, Inouye M. 2019. Elevated serum α-1 antitrypsin is a major component of GlycA-associated risk for future morbidity and mortality. PLoS One 14:e0223692.

Ritchie SC, Lambert SA, Arnold M, Teo SM, Lim S, Scepanovic P, Marten J, Zahid S, Chaffin M, Liu Y, Abraham G, Ouwehand WH, Roberts DJ, Watkins NA, Drew BG, Calkin AC, Di Angelantonio E, Soranzo N, Burgess S, Chapman M, Kathiresan S, Khera AV, Danesh J, Butterworth AS, Inouye M. 2021a. Integrative analysis of the plasma proteome and polygenic risk of cardiometabolic diseases. Nat Metab 3:1476-1483.

Ritchie SC, Surendran P, Karthikeyan S, Lambert SA, Bolton T, Pennells L, Danesh J, Di Angelantonio E, Butterworth AS, Inouye M. 2021b. Quality control and removal of technical variation of NMR metabolic biomarker data in ∼120,000 UK Biobank participants. medRxiv. doi:10.1101/2021.09.24.21264079

Sliz E, Kettunen J, Holmes MV, Williams CO, Boachie C, Wang Q, Männikkö M, Sebert S, Walters R, Lin K, Millwood IY, Clarke R, Li L, Rankin N, Welsh P, Delles C, Jukema JW, Trompet S, Ford I, Perola M, Salomaa V, Järvelin M-R, Chen Z, Lawlor DA, Ala-Korpela M, Danesh J, Davey Smith G, Sattar N, Butterworth A, Würtz P. 2018. Metabolomic consequences of genetic inhibition of PCSK9 compared with statin treatment. Circulation 138:2499-2512.

Wand H, Lambert SA, Tamburro C, Iacocca MA, O'Sullivan JW, Sillari C, Kullo IJ, Rowley R, Dron JS, Brockman D, Venner E, McCarthy MI, Antoniou AC, Easton DF, Hegele RA, Khera AV, Chatterjee N, Kooperberg C, Edwards K, Vlessis K, Kinnear K, Danesh JN, Parkinson H, Ramos EM, Roberts MC, Ormond KE, Khoury MJ, Janssens ACJW, Goddard KAB, Kraft P, MacArthur JAL, Inouye M, Wojcik GL. 2021. Improving reporting standards for polygenic scores in risk prediction studies. Nature 591:211-219.

Wray NR, Yang J, Hayes BJ, Price AL, Goddard ME, Visscher PM. 2013. Pitfalls of predicting complex traits from SNPs. Nat Rev Genet 14:507-515.

Zheng J, Haberland V, Baird D, Walker V, Haycock PC, Hurle MR, Gutteridge A, Erola P, Liu Y, Luo S, Robinson J, Richardson TG, Staley JR, Elsworth B, Burgess S, Sun BB, Danesh J, Runz H, Maranville JC, Martin HM, Yarmolinsky J, Laurin C, Holmes MV, Liu JZ, Estrada K, Santos R, McCarthy L, Waterworth D, Nelson MR, Davey Smith G, Butterworth AS, Hemani G, Scott RA, Gaunt TR. 2020. Phenome-wide Mendelian randomization mapping the influence of the plasma proteome on complex diseases. Nat Genet 52:1122-1131.

*Reviewer #3 (Recommendations for the authors):*

I have the following more specific comments that could possibly improve the study by Fang et al.:

1. The NMR platform used by the authors measures only a small number of small molecules and the vast majority of the derived measures refer to characteristics of lipoprotein particles, which are not 'classical' metabolites. The paper would benefit from a paper distinction between both measures. Therefore, it is questionable how many of the 100,000 metabolites mentioned by the authors are captured by the technology used and further it is even of interest how many approximately independent features are indeed captured by the technology. A principal component analysis or similar dimension reduction techniques would provide an important correction/estimate of/to the metabolic space captured. Given the high correlation among the NMR traits, it would be important to state how many likely independent associations have been found, for instance by clumping highly correlated metabolites in clusters, similar as the authors do for SNPs.

2. I find the section about collider bias in the introduction a bit as surprise and it is unclear to me, how this relates to the overall aim of the study. The high amount of self-citation in this section and in general throughout the paper makes me wonder, how much of an issue this really is compared to more substantial questions about the suitability of PRS to identify disease biomarkers or causal metabolites.

3. Investigating different types of genetic scores is a clear strength of this work. However, the study currently stops early leaving important questions unanswered. For instance, how many more 'helpful' associations between PRS and NMR measures are really gained by going genome-wide, that is, how many of the added associations are mainly due to unspecific pleiotropy? The authors have outstanding methodological skills in MR and related causal inference techniques, and I find it somewhat wasted here to go simply for more associations instead of digging into the relevant part, which in turn defines the usefulness of the whole atlas and hence this study. It currently reads mostly like a computational exercise.

4. In my opinion, the key question is, what do all those associations deliver, do they bring us any closer to causality compared to 'classical' observational associations, given LD confounding and strong metabolite variants included in PRS driving the associations. For example, how robust are PRS associations to the exclusion of single regions, Ritchie et al., 2021 Nat Metab provide a neat framework to address such questions. The APOE example goes in that direction and other locus-specific effects are likely underlying other disease PRS – NMR measure associations.

5. One might argue that the CKD example is somewhat self-fulling, given that the selection of cases is mainly done based on serum creatinine (or the eGFR derived from creatinine), but is a good example how strong and specific examples might point to disease biomarkers. Are there more examples for a given NMR trait being strongly and possibly specifically associated with a trait or a cluster of highly related traits? I would also tone the creatinine example done, since it is obvious that this marker is used for clinical decision making.

6. I am not quite sure about the bidirectional MR approach. By only testing diseases for which the PRS showed an association with the metabolite of interest, isn't it quite likely that an effect of the metabolite on the disease would be missed, as one would assume that a true causal association between a metabolite and a disease might well be lost in the large set of SNPs associated with the endpoint of interest.

7. I disagree with the statement on page 12 that all lipid traits associated with the CHD PRS are causal risk factors, most of them are likely not and the true underlying risk factor or biological mechanism is likely hidden among all those associations. The general inability to distinguish about the causal relevance of all those highly related measures is a massive challenge working with NMR data, in particular given that many are not real measurements but are only derived as proportion from other measures, including apolipoprotein B.

8. The authors need to do better in distinguishing between all the different lipid measures that are derived from the NMR platform, this is also seen in the discussion of Apo B. Apo B is the main structural protein for many lipoprotein particles of different densities and not just for atherogenic ones.

9. Instead of stratified analysis, why not correcting for statin intake to estimate lipid levels of participants without the use of statins? What about the effect of many other medications widely prescribed, with strong effects on NMR measures as described in van Duijn et al., Nat Med 2020?

---

## [Author Response]

Essential revisions:1) While this work provides a potentially important resource for the community, the examples provided tend to be illustrative, and do not really do justice to the resource. We strongly encourage the authors to revisit the data analysis for more-compelling examples of the types of impactful work that can be achieved using this resource. How and why should users want to use this Atlas?

Many thanks for your comments to help us refine this resource and manuscript. We have now revisited the examples in this manuscript to showcase the value of the PGS atlas with the help of the reviewer comments:

1. We have undertaken a new application of bi-directional Mendelian randomization (MR) to demonstrate how readers may use this approach to disentangle whether associations in our atlas likely reflect either causes or consequences of PGS traits/diseases. This example is described on page 9 and discussed in the discussion on page 21:

‘We likewise conducted bi-directional MR to demonstrate that associations between the CHD PGS and VLDL particle size likely reflect an effect of CHD liability on this metabolic trait. In contrast, the association between the CHD PGS and VLDL concentrations are likely attributed to the causal influence of this metabolic trait on CHD risk, suggesting that it is the concentration of these triglyceride-rich particles that are important in terms of the aetiology of CHD risk as opposed to their actual size. We envisage that findings from our atlas, as well as other ongoing efforts which leverage the large-scale NMR data within UKB, should facilitate further granular insight into lipoprotein lipid biology.’

2. Based on reviewer #2’s comments regarding our previous example using results for GlycA, we have performed further sensitivity analyses to support the conclusions from this example.

3. Based on reviewer #3’s comments we have included further explanation regarding collider bias as well as a Figure (page 14) to emphasize the importance of age-stratified analyses in our atlas for users to evaluate the robustness of findings.

4. Finally, we have also recalculated all PGS-metabolite associations in the atlas after removing established lipid loci across the genome based on the publication by Gallois et al., (2019 Nature Communications https://www.nature.com/articles/s41467-019-12703-7). As an example of why this sensitivity analysis may be helpful to the user, we compared metabolomic profiles for the anthropometric and the Alzheimer’s disease PGS before and after removing these loci (page 19):

‘For PGS with these lipid loci excluded, anthropometric traits such as waist-to-hip ratio (N=209), waist circumference (N=206) and body mass index (N=205) still provided strong evidence of association with the majority of metabolic measurements on the NMR panel based on multiple testing corrections. Elsewhere however, the Alzheimer’s disease PGS, which was associated with 60 metabolic traits robust to P<0.05/19 in the initial analysis including these lipid loci (Supplementary Table 17), provided no convincing evidence of association with the 249 circulating metabolites after excluding the lipid loci based on the same multiple testing threshold (Supplementary Table 18). Further inspection suggested that the likely explanation for this were variants located at the *APOE* locus which are recognised to exert their influence on phenotypic traits via horizontally pleiotropic pathways (Ferguson et al., 2020).’

2) One key question is whether these analyses bring us closer to causality compared to 'classical' observational associations, given LD confounding and strong metabolite variants included in PRS driving the associations. For example, how robust are PRS associations to the exclusion of individual regions?

Thank you for the suggestion. As recommended by reviewer #2 we have looked into the analysis conducted in the Ritchie et al., 2021 Nat Metab paper. The authors examined the association between 5 different genome-wide polygenic scores (PGSs) and systematically removed 1,703 approximately independent LD blocks with replacement in their association analyses.

We agree that it is very important to consider the contribution of individual gene regions when examining genetic associations. Whilst this was feasible for 5 different PGS in the study by Ritchie et al., this becomes a lot more computational intensive for our study whereby we construct 125 different PGS using 2 different selection criteria. Therefore, we have added a new analysis to evaluate the contribution of 14 previously discovered lipid-associated genes (LIPC, *APOA5*, *CETP*, *PCSK9*, *LDLR*, *GCKR*, *APOC1*, *LPL*, *GALNT2*, *CELSR2*, *TRIB1*, *DOCK7*, *FADS2* and *APOE*) in the PGS-metabolites associations as reported on page 19:

‘The polygenic nature of complex traits means that the inclusion of highly weighted pleiotropic genetic variants in PGS may introduce bias into genetic associations within our atlas. To provide insight into this issue, we constructed PGS excluding variants within the regions of the genome which encode the genes for 14 major regulators of NMR lipoprotein lipids signals which captured 75% of the gene-metabolite associations in the Finnish Metabolic Syndrome In Men (METSIM) cohort (Gallois et al., 2019). For details of these genes (see Supplementary Table 5).’

Followed by the example described above to help demonstrate to users to importance of these results for further evaluate the robustness of findings.

Second, can we determine whether the associations are causally upstream, or downstream of the identified phenotypes? At minimum the authors should demonstrate for examples how these issues can be teased apart (and optionally provide analyses atlas-wide).

As described above we have now added an example evaluating associations with measures of triglyceride-rich VLDL captured by the NMR panel by applying bi-directional MR to disentangle whether they may reflect causes or consequences of traits/disease.

The main reason for not conducting these types of evaluations atlas-wide is because we present this resource as one to generate hypotheses which users can then conduct in-depth evaluations of in a timely and careful manner. This was highlighted by reviewer #2 who notes that challenges of appropriately instrumenting the metabolic trait GlycA which we have also address below in response to their comment. This overall point has been discussed on page 8 of the manuscript:

‘In this paper, we provide several examples of how results from this atlas can be used to generate hypotheses and pave the way for in-depth and careful evaluations of associations between PGS and circulating traits. Specifically, we believe our findings can facilitate a ‘reverse gear Mendelian randomization’ approach to disentangle whether associations likely reflect metabolic traits acting as a cause or consequence of disease risk (Holmes and Davey Smith, 2019) as illustrated using triglyceride-rich very low density lipoprotein (VLDL) particles in the next section. Furthermore, in-depth evaluations allow careful consideration of appropriate instrumental variables for circulating metabolites which can be a challenging task as highlighted in our exemplar analysis of glycoprotein acetyls. Finally, we provide examples of how the plethora of sensitivity analyses within our atlas can help users further investigate the robustness of findings.’

3) The atlas would also be potentially strengthened by including additional results for sex-stratified models. Previous work has shown significant differences in NMR biomarker concentrations between males and females, so it would be valuable to see if these result in any differences in PRS associations which may arise despite PRSs largely being constructed from autosomal GWAS.

Many thanks for this suggestion. We have undertaken extensive analyses to generate all PGS-metabolite association stratified by sex and added these to the web platform. This has been discussed on page 8:

‘Our atlas also includes sex-stratified estimates for PGS weighted by GWAS undertaken in female only (such as breast cancer and age at menarche) and male only (e.g., prostate cancer) populations, as well as sex-stratified estimates in both females and males separately for all other PGS-metabolite associations. We encourage users interested in these sex-stratified estimates to interpret them with caution however, given the widespread sex-differential participation bias in UKB (Pirastu et al., (2021)).’

And page 20:

‘Moreover, we suggest that users interested in the sex-stratified estimates within our atlas should interpret them in conjunction with estimates derived across age tertiles as in this example, given that different proportions of males and females in UKB may be taking certain medications (e.g. statins).’

Furthermore, the reviewers provide numerous additional specific and insightful comments; we strongly encourage the authors to consider these points seriously in their revisions.Reviewer #1 (Recommendations for the authors):Specific comments:Some of the text on figures is very small and it may be worth revisiting figure design.

Many thanks for this suggestion. We have now regenerated our large Circos plot figure where text was particularly small (now as a forest plot) and moved it to the supplementary Figure. Furthermore, we note that this work has been submitted to *eLife* which is an online only journal, therefore meaning that readers will have the added benefit of always being able to zoom in of the figures generated in our manuscript.

P8: the positive control of creatinine and kidney disease seems a bit too trivial, given that creatinine is a diagnostic criterion. Perhaps there may be a better example?

The creatinine-CKD example was originally included to demonstrate the predictive ability of this PGS; however, this has now been removed as per your suggestion. Instead, we have further highlighted associations between lipoprotein lipid metabolic traits and the coronary heart disease (CHD) PGS, which we feel is powerful positive control given the overwhelming evidence supporting a causal relationship between lipids and CHD risk.

P7 Para1: It would be helpful to the reader to say a bit more about the range of types of traits considered here, as well as typical sample sizes given that the data come from a variety of sources.

We have now added a pie chart to illustrate the range of PGS traits/disease in our atlas into the Figure 1. A similar figure reflecting associations across NMR subcategories has also been generated alongside and added to Figure 1. All sample sizes are reported in Supplementary Table 1.

Consider using PGS instead of PRS as more accurate for quantitative and non-disease traits.

Thank you – we have now changed ‘PRS’ to ‘PGS’ throughout.

Reviewer #2 (Recommendations for the authors):Introduction/discussion: this is not the first study that has demonstrated the utility of PRS associations for prioritising molecular traits for follow-up. We have recently published a paper doing so for cardiometabolic PRS and protein levels (Ritchie et al., 2021a) and this has been available as a preprint in various forms since 2019. This should be noted and cited appropriately at the relevant points in the introduction and discussion.

We have now referenced your paper on pages 4 and 12 of the manuscript.

The associations in the atlas for the systolic blood pressure (SBP) and diastolic blood pressure (DBP) are invalid. As the authors note in the methods, PRSs for these two traits are constructed from GWAS that include UK Biobank samples. PRSs are invalid when used in samples that contributed to their underlying GWAS (Lambert et al., 2021; Wand et al., 2021; Wray et al., 2013) as this causes them to have dramatically inflated associations. Associations for these two PRS should be removed from the atlas and study, or if the authors think these two traits are of critical interest, an alternate source of GWAS that do not include UK Biobank participants should be used to construct these two PRSs.

Many thanks for your suggestion. We have now removed the SBP and DBP PGS from our atlas due to overlapping samples in UKB.

We have found significant sources of technical variation exist in the NMR metabolomics data for UK Biobank (Ritchie et al., 2021b), which may confound some of the PRS associations here. Of particular concern in the context of this study are the presence of outlier shipping plates, on which samples have systematically high (or low) concentrations of non-biological origin (see Figure 5 in Ritchie et al., 2021b). This affects up to 5% of samples depending on the biomarker, but will have an outsized effect on associations as they result in samples being spuriously located at the extremities of biomarker distributions. I.e. this may result in associations being weaker or not significant, or less likely, false positives may be introduced if outlier plates happen to correlate with PRS. There are also a small number of biomarkers significantly impacted by other sources of technical variation, e.g. drift over time within spectrometer, or sample degradation due to extended time between sample preparation and sample measurement. We have made available an R package: ukbnmr (https://github.com/sritchie73/ukbnmr), for correcting for this technical variation, and removing samples on these outlier plates. Although this preprint is still under review, we would suggest using this package to remove the described technical variation, or at least checking whether doing so significantly changes associations in the atlas.

Thank you for this suggestion. As mentioned in response to the public review, we have now repeated our entire pipeline after applying your R package.

There are also major biological determinants of NMR biomarker concentrations that have not been accounted for, and thus may confound, PRS to biomarker associations. In addition to age, sex, and 10 genetic principal components that the authors already adjust for, NMR biomarker concentrations are also strongly correlated with body mass index (BMI), fasting time, and lipid lowering medication (statin) usage (see Figure 8 in Ritchie et al., 2021b). The atlas would be greatly improved by either appropriately accounting for these in the basic models, or including additional results using models that do so.Regarding statin usage in particular, we appreciate that the authors have conducted a separate age-stratified analysis to evaluate the impact of medication usage on PRS to biomarker associations. However, the fact remains that statin usage will be confounding the PRS to biomarker associations in the main results (e.g. by artificially lowering LDL cholesterol in people at high CHD PRS). This confounding is typically removed in one of two ways: either (1) fitting associations excluding participants taking statins, or (2) applying biomarker-specific correction factors to concentrations prior to association analysis, such as those estimated by (Kofink et al., 2017; Sliz et al., 2018). The latter is probably the better approach as the former is likely to introduce further bias towards young and healthy participants.

We have responded to this suggestion in the public review. However, just to add that the previous studies referenced here are noticeably much smaller in sample size compared to our study (n=864 & n=5359 respectively). Conditioning on colliders has become a lot more problematic with the advent of biobank scale cohort studies such as UKB. As such, previously modest sources of bias induced by conditioning on colliders, which may have not led to spurious findings in smaller cohort studies, are now large enough such that precautions (e.g., those taken in our study by stratifying on age) are necessary to investigate the robustness of findings.

The atlas would also be potentially strengthened by including additional results for sex-stratified models. We have observed significant differences in NMR biomarker concentrations between males and females, so it would be interesting to see if these result in any differences in PRS associations which may arise despite PRSs largely being constructed from autosomal GWAS.

We have undertaken extensive analyses to additionally evaluate sex-stratified associations (page 8):

‘Our atlas also includes sex-stratified estimates for PGS weighted by GWAS undertaken in female only (such as breast cancer and age at menarche) and male only (e.g., prostate cancer) populations, as well as sex-stratified estimates in both females and males separately for all other PGS-metabolite associations. We encourage users interested in these sex-stratified estimates to interpret them with caution however, given the widespread sex-differential participation bias in UKB (Pirastu et al., (2021)).’

And page 20:

‘Moreover, we suggest that users interested in the sex-stratified estimates within our atlas should interpret them in conjunction with estimates derived across age tertiles as in this example, given that different proportions of males and females in UKB may be taking certain medications (e.g. statins).’

Regarding the bi-directional Mendelian randomization analysis of GlycA, we would suggest using an alternative biomarker as exemplar in the study, as GlycA is heterogeneous and needs to be more carefully instrumented than has been done in the study currently. GlycA is an NMR signal that quantifies the total concentrations of five proteins in circulation (Otvos et al., 2015): α-1-acid glycoprotein (AGP), α-1-antitrypsin (AAT), α-1-antichymotrypsin (AACT), haptoglobin (HP), and transferrin (TF). These are acute-phase reactants whose concentrations each change in response to acute inflammation or in chronic inflammation (Connelly et al., 2017). Moreover these changes are differential with respect to each other, and over time (Ebersole and Cappelli, 2000; Gabay and Kushner, 1999). Further, a single GlycA measurement cannot be simply decomposed as two people with the same GlycA concentration can have different concentrations of the underlying proteins (Ritchie et al., 2019). This heterogeneity means that using genome-wide significant QTLs for GlycA is unlikely to be appropriate as these QTLs may include (1) protein-QTLs for one or more of the five proteins GlycA captures, and (2) signals involved in initiation of acute-phase response (e.g. interleukin-6 signalling pathways). To use GlycA as an exposure in Mendelian randomization analysis, the QTLs selected as instruments should be restricted to cis-pQTLs for the five proteins. These signals may need to be treated separately, as in the scenario GlycA is truly causal, it is unlikely that all five proteins are causal, or that they have similar causal effect sizes.

Thank you for the suggestion. We have repeated the GlycA Mendelian randomization analysis using the pQTLs (for AAT, AACT, HP and TF) and eQTL (for AGP) of these proteins. Detailed methods are presented on page 26 of the manuscript:

‘The NMR signal of serum GlycA is contributed by five acute-phase proteins (alpha1-acid glycoprotein, haptoglobin, alpha1-antitrypsin, alpha1-antichymotrypsin, and transferrin) (Otvos *et al.*, 2015). Thus, another set of genetic instruments for GlycA were selected among SNPs strongly associated with these five proteins for further evaluation of the genetically predicted effect of GlycA. Instrumental variables for alpha1-acid glycoprotein are identified from the gene expression quantitative trait loci (eQTL) (P<5x10^-8^, r^2^<0.001) of ORM1 from the eQTLGen Consortium (Võsa *et al.*, 2021). Genetic instruments for haptoglobin, alpha1-antitrypsin, alpha1-antichymotrypsin, and transferrin are their protein quantitative trait loci (pQTL) (P<5x10^-8^, r^2^<0.001) identified from 35,559 Icelanders (Ferkingstad *et al.*, 2021). To identify cis-acting genetic variants, genetic variants of the five proteins are limited to those located within 1Mb around their encoding genes: ORM1 (encoding alpha1-acid glycoprotein; Ensembl ID: ENSG00000229314), HP (encoding haptoglobin; Ensembl ID: ENSG00000257017), SERPINA1 (encoding alpha1-antitrypsin; Ensembl ID: ENSG00000197249), SERPINA3 (encoding alpha1-antichymotrypsin; Ensembl ID: ENSG00000196136) or TF (encoding transferrin; Ensembl ID: ENSG00000196136). GWAS summary statistics of the protein data were performed using an in-house reference panel consisting of 10,000 randomly selected European participants of the UK Biobank (Kibinge *et al.*, 2020) in PLINK v1.90. The eQTL data from the eQTLGene Consortium was extracted from the OpenGWAS portal.’

Findings can be found on page 11 of the manuscript:

‘We also conducted further sensitivity analyses given that the NMR signal of GlycA is a composite signal contributed by the glycan *N*-acetylglucosamine residues on five acute-phase proteins, including alpha1-acid glycoprotein, haptoglobin, alpha1-antitrypsin, alpha1-antichymotrypsin, and transferrin (Otvos *et al.*, 2015). Using cis-acting plasma protein (where possible) and expression quantitative trait loci (pQTLs and eQTLs) as instrumental variables for these proteins (Supplementary Table 12) did not provide convincing evidence that they play a role in disease risk for associations between PGS and GlycA (Supplementary Table 13). The only effect estimate robust to multiple testing was found for higher genetically predicted alpha1-antitrypsin levels on γ glutamyl transferase (GGT) levels (Β=0.05 SD change in GGT per 1 SD increase in protein levels, 95% CI=0.03 to 0.07, FDR=3.6x10^-3^), although this was not replicated when using estimates of genetic associations with GGT levels from a larger GWAS conducted in the UK Biobank data (Β=1.6x10^-3^, 95% CI=-6.9 x10^-3^ to 0.01, P = 0.71). For details of pleiotropy robust analysis and replication results see Supplementary Table 14.’

Data availability: the PRS should also be made publicly available, e.g. downloadable via the atlas. These should include at minimum: the set of variants in each PRS, variant identifier information (e.g. chromosome and position), genome build, effect allele, and weight (β or log odds from the underlying GWAS).

All weights used to generate the PGS in our atlas are now available at https://tinyurl.com/PGSweights as mentioned on page 7:

‘The specific weights for clumped variants used in all PGS can be found at https://tinyurl.com/PGSweights.’

Reviewer #3 (Recommendations for the authors):Apart from the general comments made above, I have the following more specific comments that could possibly improve the study by Fang et al.:1. The NMR platform used by the authors measures only a small number of small molecules and the vast majority of the derived measures refer to characteristics of lipoprotein particles, which are not 'classical' metabolites. The paper would benefit from a paper distinction between both measures. Therefore, it is questionable how many of the 100,000 metabolites mentioned by the authors are captured by the technology used and further it is even of interest how many approximately independent features are indeed captured by the technology. A principal component analysis or similar dimension reduction techniques would provide an important correction/estimate of/to the metabolic space captured. Given the high correlation among the NMR traits, it would be important to state how many likely independent associations have been found, for instance by clumping highly correlated metabolites in clusters, similar as the authors do for SNPs.

As suggested we have conducted a PCA analysis to investigate the amount of metabolic space captured. Findings are reported on page 8:

‘Principal component analysis suggested that the first 19 principal components (PCs) captured 95% of the variance in the NMR metabolites data (whereas the first ten PCs captured 90% and the first 41 PCs captured 99% of the variance) (Supplementary Table 3).’

The full results from PCA analysis are presented in Supplementary Table 3.

2. I find the section about collider bias in the introduction a bit as surprise and it is unclear to me, how this relates to the overall aim of the study. The high amount of self-citation in this section and in general throughout the paper makes me wonder, how much of an issue this really is compared to more substantial questions about the suitability of PRS to identify disease biomarkers or causal metabolites.

We have now added a more detailed section (including a new figure in the main document) discussing the challenges of addressing potential collider in a biobank-scale dataset which is discussed on page 14:

‘Directed acyclic graphs illustrating the potential collider bias involved in the causal relationship between the coronary artery disease polygenic score and circulating metabolites. (A) The likelihood of participants in UK Biobank taking medication such as statins is influenced by having a higher genetic predisposition to coronary artery disease but may also be influenced by certain metabolic traits measured on the NMR panel (e.g. having elevated low-density lipoprotein cholesterol levels). Either stratifying or adjusting for statin use in regression models may therefore induce collider bias into the association between disease liability and metabolic traits. (B) Age is commonly adjusted for in association analyses due to its role as a confounder and cannot be a collider (i.e. exposures and outcomes cannot influence the age of participants). Stratifying samples by age therefore enables the analysis of exposure-outcome associations in a group of participants with relatively consistent confounding effect from age, leading to more robust association estimates in the lower age tertile where the percentage of participants who are regularly taking medication is low. Furthermore, comparisons with participants in the highest age tertile can help highlight associations between polygenic scores and metabolic traits most likely distorted by potential colliders such as statins in the full sample.’

3. Investigating different types of genetic scores is a clear strength of this work. However, the study currently stops early leaving important questions unanswered. For instance, how many more 'helpful' associations between PRS and NMR measures are really gained by going genome-wide, that is, how many of the added associations are mainly due to unspecific pleiotropy? The authors have outstanding methodological skills in MR and related causal inference techniques, and I find it somewhat wasted here to go simply for more associations instead of digging into the relevant part, which in turn defines the usefulness of the whole atlas and hence this study. It currently reads mostly like a computational exercise.

Thank you for your complement re. methodological skills. We have emphasized in our introduction (page 8) that the main purpose of this resource is to generate hypotheses and therefore facilitate in-depth, carefully designed follow-up analyses (such as those applying the principles of MR). We advocate in this section that causal inference is challenging to robustly undertake which is why we suggest readers conduct bespoke analyses for this task motivated by findings in this atlas of results:

‘In this paper, we provide several examples of how results from this atlas can be used to generate hypotheses and pave the way for in-depth and careful evaluations of associations between PGS and circulating traits. Specifically, we believe our findings can facilitate a ‘reverse gear Mendelian randomization’ approach to disentangle whether associations likely reflect metabolic traits acting as a cause or consequence of disease risk (Holmes and Davey Smith, 2019) as illustrated using triglyceride-rich very low density lipoprotein (VLDL) particles in the next section. Furthermore, in-depth evaluations allow careful consideration of appropriate instrumental variables for circulating metabolites which can be a challenging task as highlighted in our exemplar analysis of glycoprotein acetyls. Finally, we provide examples of how the plethora of sensitivity analyses within our atlas can help users further investigate the robustness of findings.’

4. In my opinion, the key question is, what do all those associations deliver, do they bring us any closer to causality compared to 'classical' observational associations, given LD confounding and strong metabolite variants included in PRS driving the associations. For example, how robust are PRS associations to the exclusion of single regions, Ritchie et al., 2021 Nat Metab provide a neat framework to address such questions. The APOE example goes in that direction and other locus-specific effects are likely underlying other disease PRS – NMR measure associations.

We have undertaken extensive sensitivity analyses to compute all PGS association in this atlas after removing established lipid gene loci to evaluate the sensitivity of results driven by variants at these regions of the genome. This has been added to page 19 of the manuscript:

‘The polygenic nature of complex traits means that the inclusion of highly weighted pleiotropic genetic variants in PGS may introduce bias into genetic associations within our atlas. To provide insight into this issue, we constructed PGS excluding variants within the regions of the genome which encode the genes for 14 major regulators of NMR lipoprotein lipids signals which captured 75% of the gene-metabolite associations in the Finnish Metabolic Syndrome In Men (METSIM) cohort (Gallois *et al.*, 2019). For details of these genes (see Supplementary Table 5).

For PGS with these lipid loci excluded, anthropometric traits such as waist-to-hip ratio (N=209), waist circumference (N=206) and body mass index (N=205) still provided strong evidence of association with the majority of metabolic measurements on the NMR panel based on multiple testing corrections. Elsewhere however, the Alzheimer’s disease PGS, which was associated with 60 metabolic traits robust to P<0.05/19 in the initial analysis including these lipid loci (Supplementary Table 17), provided no convincing evidence of association with the 249 circulating metabolites after excluding the lipid loci based on the same multiple testing threshold (Supplementary Table 18). Further inspection suggested that the likely explanation for this attenuation of evidence were due to variants located within the *APOE* locus which are recognised to exert their influence on phenotypic traits via horizontally pleiotropic pathways (Ferguson *et al.*, 2020).’

5. One might argue that the CKD example is somewhat self-fulling, given that the selection of cases is mainly done based on serum creatinine (or the eGFR derived from creatinine), but is a good example how strong and specific examples might point to disease biomarkers. Are there more examples for a given NMR trait being strongly and possibly specifically associated with a trait or a cluster of highly related traits? I would also tone the creatinine example done, since it is obvious that this marker is used for clinical decision making.

We have now removed the creatinine example and instead included a bi-directional MR analysis of VLDL metabolic traits (page 9).

‘For example, we applied Mendelian randomization (MR) to further evaluate associations highlighted in our atlas with triglyceride-rich very-low-density lipoprotein (VLDL) particles. For instance, both VLDL particle average diameter size and concentration were associated with the PGS for body mass index (BMI) (Β=0.04, 95% CI=0.033 to 0.046, P=3.53x10^-35^ & Β=0.012, 95% CI=0.006 to 0.019, P=2.7x10^-4^ respectively) and coronary heart disease (CHD) (Β=0.026, 95% CI=0.019 to 0.032, P< 2.12x10^-15^ & Β=0.035, 95% CI=0.028 to 0.042, P< 2.73x10^-24^ respectively). Conducting bi-directional MR suggested that the associations with average diameter of VLDL particles are likely attributed to a consequence of BMI and CHD liability as opposed to the size of VLDL particles having a causal influence on these outcomes (Supplementary Table 6). In contrast, MR analyses suggested that the concentration of VLDL particles increases risk of CHD (Β=1.28 per 1-SD change in VLDL particle concentration, 95% CI=1.25 to 1.65, P=2.8x10^-7^) which may explain associations between the CHD PGS and this metabolic trait within our atlas. Similar MR analyses to investigate findings from our atlas can be conducted using the full GWAS summary statistics for all 249 circulating metabolic traits available via the GWAS catalog (https://www.ebi.ac.uk/gwas/) under accession IDs GCST90092803 to GCST90093051.’

6. I am not quite sure about the bidirectional MR approach. By only testing diseases for which the PRS showed an association with the metabolite of interest, isn't it quite likely that an effect of the metabolite on the disease would be missed, as one would assume that a true causal association between a metabolite and a disease might well be lost in the large set of SNPs associated with the endpoint of interest.

The value of using PGS with a lenient threshold is that associations may highlight either causes or consequences of disease risk. This has been previously referred to as ‘reverse gear MR’ as proposed in the study by Holmes and Davey Smith (https://academic.oup.com/clinchem/article-lookup/doi/10.1373/clinchem.2018.296806). This was one of the primary motivations for choosing the VLDL example mentioned above as well as providing insight into lipoprotein lipid biology (page 21):

‘We likewise conducted bi-directional MR to demonstrate that associations between the CHD PGS and VLDL particle size likely reflect an effect of CHD liability on this metabolic trait. In contrast, the association between the CHD PGS and VLDL concentrations are likely attributed to the causal influence of this metabolic trait on CHD risk, suggesting that it is the concentration of these triglyceride-rich particles that are important in terms of the aetiology of CHD risk as opposed to their actual size. We envisage that findings from our atlas, as well as other ongoing efforts which leverage the large-scale NMR data within UKB, should facilitate further granular insight into lipoprotein lipid biology.’

7. I disagree with the statement on page 12 that all lipid traits associated with the CHD PRS are causal risk factors, most of them are likely not and the true underlying risk factor or biological mechanism is likely hidden among all those associations. The general inability to distinguish about the causal relevance of all those highly related measures is a massive challenge working with NMR data, in particular given that many are not real measurements but are only derived as proportion from other measures, including apolipoprotein B.

We have now rephrased this sentence (now on page 15) to address this comment:

‘The vast majority of these were lipoprotein lipid traits, which are likely capturing causal risk factors for CHD.’

8. The authors need to do better in distinguishing between all the different lipid measures that are derived from the NMR platform, this is also seen in the discussion of Apo B. Apo B is the main structural protein for many lipoprotein particles of different densities and not just for atherogenic ones.

We agree that it is important to distinguish between the various metabolic traits measured by the NMR platform. Given that this work is a resource paper, we have pointed readers to a review of this platform to help them navigate findings within our atlas (page 4):

‘The 249 metabolite measurements include the particle concentration, size, and composition of 14 lipoprotein subclasses, as well as the levels of phospholipids, fatty acids, amino acids, ketone bodies and others biomarkers as discussed in a recent review (Ala-Korpela *et al.*, 2022).’

9. Instead of stratified analysis, why not correcting for statin intake to estimate lipid levels of participants without the use of statins? What about the effect of many other medications widely prescribed, with strong effects on NMR measures as described in van Duijn et al., Nat Med 2020?

As discussed above, we have added a section to page 14 to clarify our justification of age-stratified analyses to evaluate the robustness of findings to potential colliders such as statin therapy. Conditioning on colliders (including the use of statins) has become a lot more problematic with the advent of biobank scale cohort studies such as UKB. As such, previously modest sources of bias induced by conditioning on colliders, which may have not led to spurious findings in smaller cohort studies such as the one referenced here by the reviewer, are now large enough such that precautions (e.g. those taken in our study by stratifying on age) are necessary to investigate the robustness of findings.